# HPV-Vaccine Hesitancy in Colombia: A Mixed-Methods Study

**DOI:** 10.3390/vaccines10081187

**Published:** 2022-07-27

**Authors:** Veronica Cordoba-Sanchez, Mariantonia Lemos, Diego Alfredo Tamayo-Lopera, Sherri Sheinfeld Gorin

**Affiliations:** 1Department of Psychology, School of Social Sciences, Institucion Universitaria de Envigado, Envigado 055422, Colombia; datamayo@correo.iue.edu.co; 2Department of Psychology, School of Arts and Social Sciences, Universidad EAFIT, Medellín 050022, Colombia; mlemosh@eafit.edu.co; 3School of Medicine, University of Michigan, Ann Arbor, MI 48109, USA

**Keywords:** vaccination, human papilloma virus, health behaviour, vaccine hesitancy, cervical cancer

## Abstract

In Colombia, the uptake rate of the HPV vaccine dropped from 96.7% after its introduction in 2013 to 9% in 2020. To identify the behavioural components of HPV-vaccine hesitancy in females aged 15 and under and their families, we conducted a convergent mixed-methods study in which 196 parents/caregivers responded to an online questionnaire and 10 focus groups were held with 13 of these parents/caregivers, and 50 age-eligible girls. The study is novel as it is the first to explore the factors influencing HPV-vaccine hesitancy alongside the COVID vaccine within an integrative model of behaviour change, the capability-opportunity-motivation-behaviour (COM-B) model. We found that COVID-19 has had an impact on the awareness of HPV and HPV vaccination. Lack of information about the vaccination programs, concerns about vaccine safety and the relationship between HPV and sexuality could be related to vaccine hesitancy. Trust in medical recommendations and campaigns focused on the idea that vaccination is a way of protecting daughters from cervical cancer could improve HPV vaccine uptake.

## 1. Introduction

Cervical cancer is the fourth-most-common cancer among women worldwide. In 2020, there were an estimated of 604,127 new cases and 341,831 deaths [1]. In less-developed countries, its incidence is higher and it ranks second for mortality, after breast cancer [2]. In Colombia, cervical cancer has a crude incidence rate of 18.3 per 100,000 women per year (95% UI: 4.311–5.216) and a mortality rate of 9.61 (95% UI: 2.316–2.677) [3]. In 2020, 7.9% of all new cancer cases in Colombian women were of cervical cancer, the equivalent of 4742 new cases [4]. Population-based studies in this country report that women diagnosed with cervical cancer are younger, of lower incomes, and more often live in non-metropolitan/rural areas than those diagnosed with breast cancer [5]. The highest mortality rates are observed in the most deprived regions (along the main rivers, harbours, and cities along the country’s borders). The low impact of cervical cancer screening programs in the country is attributed to the poor quality of pap smears; low coverage, especially of women at high risk; and a lack of or partial follow-up of women with abnormal cytology [6].

Almost all cervical cancers are caused by persistent infections with oncogenic, or high-risk, types of human papillomavirus [7,8]. Rates of cervical cancer have declined worldwide in countries with successful HPV-vaccination strategies, cervical cancer screening programs, and the treatment of cervical intraepithelial neoplasia (CIN) earlier in the pathogenesis pathway to invasive cervical cancer [9,10,11,12,13,14,15,16,17,18,19,20,21]. By contrast, countries without cervical cancer prevention and control strategies—or where the strategies are not effective—have seen a rapid increase in early mortality due to this pathology [2], especially in low- and middle-income countries [22]. To encourage wider implementation of successful cervical cancer prevention and control strategies worldwide, on 17 November 2020, after the closure of the 73rd World Health Assembly, the WHO Global Strategy for Cervical Cancer Elimination was formalized [23]. WHO set the following goals for 2030: 90% of girls fully vaccinated with the HPV vaccine by the age of 15, 70% of women screened using a high-performance test by the age of 35, and again by the age of 45, and 90% of women diagnosed with pre-cancer and invasive cancer treated [24].

While HPV vaccines are effective against many of the oncogenic types of HPV, vaccination uptake and completion are low among young Colombian females. In 2012, Colombia was among the first countries in South America to implement HPV vaccination among age-eligible girls, reaching 97.5% uptake of first doses and 96.7% uptake of second doses. In 2013, Colombia’s HPV vaccination rate was one of the highest in the world [25]. In 2014, however, a group of young women in a Colombian coastal town who had been vaccinated experienced a mass psychogenic response including fainting spells, weakness, limb paraesthesia, chest pain, tachycardia, and headaches (the “Carmen de Bolivar” event). The purported vaccine side effects were videoed by various media outlets and shared widely on social networks [26]. While the Instituto Nacional de Salud Colombiano (Colombian National Health Institute) conducted a rigorous epidemiological study of this event and did not find any organic association between the vaccine and symptoms described [27], the lingering effects of the event continued to lower HPV vaccine rates in 2016 to 14% for first doses and 5% for the second. By 2021, however, the rate of HPV uptake had risen to 39.4% for first doses and 11.8% for second doses, according to the Colombia Health Ministry [28].

This crisis was similar to others in countries such as Denmark, Japan, and Austria. Denmark experienced a rapid decline in vaccination in 2014 following negative public attention coinciding with increased suspected adverse event reporting to the Danish Medicines Agency. This negative public attention included stories of the supposedly harmful effects of HPV vaccination that were widely shared on social media [29,30]. In Japan, active recommendations for HPV vaccination were suspended in June 2013 following media reports of girls having various symptoms such as chronic pain and motor impairment after vaccination [31]. The suspension remained until November 2021, despite large-scale epidemiologic studies showing the effectiveness and safety of the vaccine in Japan and worldwide, and the scientific community repeatedly calling for the resumption of active recommendations by the Japanese government [31]. In Austria, the HPV vaccine was licensed in 2006, comparatively early relative to other countries, but was not free of debate in a nation with a historical scepticism towards vaccination. To retain the HPV vaccine, policy makers and scientific experts disassociated the vaccine from gender, vaccine manufacturers, and youth sexuality, ultimately making the vaccine a strength of the Austrian Immunization Program [32].

These crises have led to lowering the rates of HPV vaccination worldwide, with the current rate of HPV vaccination far below the 90% goal proposed by the WHO [23]. Factors associated with vaccine hesitancy in South America are a lack of information and doubts about its safety and effectiveness [33]. These results were similar to those found in a qualitative study of young females in Colombia that highlighted that parents/caregivers of girls eligible for the first dose of the HPV vaccine had little knowledge about the aetiology of cervical cancer, had not received information about vaccination benefits, and feared its adverse events [34].

Since HPV vaccination requires parental consent, parental hesitancy may be one of the strongest influences on uptake among adolescent females [35]. Parental HPV-vaccine hesitancy is a complex phenomenon that is influenced by lack of information, the opinions of important others, the attitudes/beliefs of healthcare providers, news about adverse effects, religious beliefs, and, as HPV is a sexually-transmitted infection, attitudes towards adolescent sexual behaviour [36].

The aim of this study is to identify the behavioural components of HPV-vaccine hesitancy among girls aged 15 and under and their families at educational institutions in Colombia. These factors are best understood within an integrative model of behaviour change. This study relies on the capability-opportunity-motivation-behaviour (COM-B) model of behaviour [37]. To our knowledge, the study is the first to explore the factors influencing HPV-vaccine hesitancy alongside the COVID vaccine in Colombia within an integrative model of behaviour change.

In accord with a phased approach to behavioural research [38], the findings from this study are intended to undergird a subsequent efficacy trial of a behavioural intervention to decrease HPV-vaccine hesitancy.

## 2. Materials and Methods

### 2.1. Study Design

This study used a convergent mixed-methods design with multiple data collection strategies to ensure completeness and integration of results, and to describe the perspectives of participants at different levels in a system (Figure 1).

### 2.2. Setting

Medellín is in the Metropolitan Area of the Aburra Valley; this is the second-largest urban area of Colombia with more than four million people. The city has 229 public schools (from first to eleventh grade of basic education) and 337 private (with tuition) schools (including kindergarten only) [39]. Healthcare in Colombia is both private, or contributory, and public, with a government subsidy, depending on household income levels. The HPV-vaccine is administered in schools, hospitals, and health centres in a two-dose series (0, 6 months) with no cost for girls between 9 and 17 years old and a three-doses series (0, 6, 60 months) for women 18 years and older [40].

### 2.3. Study Model

We used the COM-B model (capability, opportunity, motivation, and behaviour [38]), to select the study’s quantitative measures and for qualitative data collection, and as a guide for analysing the data. COM-B posits that behaviour change is influenced by three factors: (1) capability, having the physical and psychological abilities to engage in the behaviour; (2) opportunity, having the physical or social opportunities to engage in the behaviour; and (3) motivation, psychological processes that energise and direct behaviour such as the belief that a behaviour is important and/or socially desirable [41].

### 2.4. Participants

All age-eligible female students (9–15 years) with an incomplete vaccination schedule (0 or 1 dose) and their parents from schools in Medellín and surrounding cities were recruited to participate in the study. The convenience sample was comprised of 196 parents who consented to participate in the online questionnaire and 50 girls who participated in the focus groups after their parents’ consent was given.

### 2.5. Study Measures

Based on the COM-B model, we developed an online questionnaire that was administered to the sample of parents (see items in Appendix A). The quantitative measures were: perceived susceptibility, vaccine awareness, trust, beliefs about vaccine safety and efficacy, likelihood of vaccinating, factors influencing the decision to vaccinate or to complete the vaccination schedule, and intention to complete the vaccination schedule.

The same questions were asked of both the parents and the young women who participated in 10 focus groups, alongside prompts to encourage them to explain their responses more fully, to gain a deeper understanding of their thinking patterns within their social contexts [42]. An example is: “If you are worried about the safety of the HPV vaccine, tell us—what are your concerns?” By matching the qualitative focus-group questions to the quantitative survey items, we were able to integrate and interpret findings from the two approaches in a joint display.

### 2.6. Procedure

A convenience sample of three public and one private school was recruited; within each school, invitations were sent to parents to complete the online questionnaire and to participate in focus groups. In addition, to reach the largest possible number of participants in the defined geographical area, the online questionnaire was also shared in the social-media and instant-messaging groups of parents from different schools. Parents/caregivers who attended the focus groups were also asked for permission for us to contact their daughters to participate in focus groups with other girls from their own school and of the same approximate age. Groups with parents/caregivers were conducted using video-call platforms (Microsoft Teams and Google Meet) and lasted an average of 34.7 min. They were coordinated by a male psychologist and two research assistants. The girls’ groups lasted an average of 28 min and were coordinated by a female psychologist and two research assistants using a room and schedule provided by the schools.

### 2.7. Ethics

Before starting, both parents and girls gave their consent and assent, respectively, in writing. All focus groups were recorded and then transcribed verbatim for subsequent analysis. Data were collected between September and November 2021. This project was approved by the Universidad EAFIT Ethics Board.

### 2.8. Data Analysis

The sample was characterized by descriptive statistics. Associations were established between categorical variables using Fisher’s chi-square, and correlations were established using Spearman’s rho. A binary logistic regression analysis of the probability of HPV vaccination was conducted, with the following independent variables: socioeconomic position, religion, factors associated with vaccination uptake such as cost, and medical recommendation, in accord with the COM-B model.

The qualitative data were analysed with thematic analysis [43], using AtlasTi version 7.5.4 for data management. Thematic analysis included: (1) Familiarization: Transcribing the entire dataset, and reading it twice to gain familiarity with the data; (2) Coding: generating initial codes, attaching codes to significant quotes, refining codes through two rounds to collate or split them if necessary; (3) Generating initial themes: examining the codes and looking for patterns and connections to cluster similar codes and create candidate themes; (4) Developing and reviewing themes, using visual maps and reading the codes of each theme; (5) Refining and naming themes; and (6) Writing the synthesis of each theme and attaching illustrative quotes.

The data were integrated by assessing the concordance between the quantitative and qualitative results according to the components of the COM-B model. As is common in mixed-methods analyses, quantitative data were elaborated and further explained with the qualitative data through a joint display table, a strategy to organize and integrate the data and show how they were mixed [44]. Our joint display showed both the quantitative factors influencing HPV-vaccine hesitancy for parents/caregivers and daughters based on the COM-B model and the qualitative information to support each component.

## 3. Results

### 3.1. Description of the Sample

One hundred and ninety-six adults, with an average age of 42.2 years (SD = 6.39), 89.3% of whom were female, responded to the online questionnaire (see Table 1). Of the respondents, 13 parents/caregivers agreed to participate in the focus groups. Fifty girls aged 9–15 (average age of 11.5 years, SD = 1.9) who were students from schools in Medellín and surrounding cities, with an incomplete vaccination schedule (0 or 1 dose), participated in the focus groups (see Table 2).

### 3.2. Analysis Based on the COM-B Model

In the sections that follow, we present both quantitative and qualitative findings aligned with the COM-B model. Qualitative themes were determined by matching content of the focus groups with constructs from the model after analysis (see Table 3). In the COM-B component capability, the themes that we established were ‘lack of information’ and ‘the relationship between HPV and cervical cancer.’ The only theme describing opportunity was trust in traditional institutions. The association of HPV with sexuality and the vaccine as an act of care was related to automatic motivation, while respect for the personal decision to be vaccinated was related to reflective motivation. One theme that was transversal to each component was the impact of COVID-19 on the conception of HPV.

#### 3.2.1. Capability

Most participants perceived a susceptibility to cervical cancer and HPV, and were aware that their daughters should be vaccinated, but 69.4% stated they had not received any information about the vaccine. Likewise, there was a relationship between the likelihood of getting their daughter vaccinated and perceived susceptibility to HPV, as well as receiving positive information about the vaccine. Qualitative analyses confirmed that parents did not have enough information about the vaccine and vaccine programs, and some cited misinformation about the Carmen de Bolivar event. Additionally, there is a recognition that cancer is a severe disease that could be treatable if it is diagnosed early. However, participants were not completely aware of the link between HPV and cancer because they thought that, given the age of the girls, they were not at risk.

##### Lack of Information

Participants reported a lack of information about HPV; further, many still remembered the images of the women and girls who fainted en masse in Carmen de Bolivar. For example, one of the parents said: “*I saw on the news when the girls from the coast were vaccinated and they fainted, so there is no one in my family who has received those vaccines*” (mother of unvaccinated girl, 45 years old, private school). The legacy of this event has been distorted, with misinformation highly prevalent. Participants stated that the correct information needs to be provided to individual parents by doctors, but this approach is limited to one parent at a time. Thus, the larger responsibility for increasing public awareness and public education rests with the health and educational systems.

“*I think that there has been a need for an education campaign, that is, promulgation of what papilloma is and what the vaccine does, because normally when a vaccine is talked about repeatedly by many media and people get educated, people have more awareness and then they do it conscientiously, but I have not heard that there has been much of a campaign or education on the subject*”.(mother of unvaccinated girl, 51 years old, private school)

For some parents, however, the first mention of the HPV vaccine is as a requirement for school: “*In my home we realized, first at school because they were asking about the vaccination card, and then with the prepaid healthcare provider; there the doctor talked about the vaccine*” (mother of girl with one dose, 41 years old, private school).

Some parents had gained limited and biased knowledge from their physicians, only in the context of their own diagnoses.

“*I was diagnosed with HPV 10 or 12 years ago (…) they explained to me that even the nuns get it [HPV], even if they haven’t had sex. I am in this meeting because I have many gaps [in my knowledge] and ignorance on the subject*”.(mother of unvaccinated girl, 45 years old, private school)

##### Relationship between HPV and Cervical Cancer

Among all the focus groups of girls, cancer meant severe disease, with difficult treatments, and death as a possible outcome. However, this was not specific to cervical cancer but to the concept of cancer. “*For me, the worst thing that can happen to a person is the treatments they have to do, because they need quick treatment or they can die, because it is a very serious cancer, and it affects women*”. (girl, one dose, 10 years old, public school).

For parents, cervical cancer is treatable when detected early. “*They do a surgery on that, don’t they? If it is detected in time, they operate on it, then people do not metastasize or something like that. That’s the only thing I know*” (mother of girl with one dose, 41 years old, private school).

Girls recognize the value of the vaccine for reducing HPV susceptibility and severity. The participants imagine (incorrectly) that the vaccine puts a little bit of the virus into the body; thus, if they get the disease, it will not be serious. They know that several doses are required for it to be effective, and one of them believed that the effect of the vaccine gets lost over time. “*They put bits of the virus into the vaccine so that when the virus reaches your body the virus doesn’t hit you so hard, so that’s what I know about vaccines*” (girl, one dose, 10 years old, public school).

#### 3.2.2. Opportunity

In focus groups of parents, participants mentioned that healthcare providers and schools were trusted information sources. Girls tended to trust their families, especially their mothers.

##### Trust in Traditional Institutions

The family and the health system are most trusted sources from which to receive information about the vaccine, and to make recommendations. For example, one of the girls said, “*If your mother tells you [to get vaccinated] it is because you mother wants the best for your life*” (unvaccinated girl, 9 years old, private school).

These two institutions are so important that they can alter previous negative attitudes and beliefs toward the vaccine. The healthcare provider can influence HPV vaccine uptake, particularly when other vaccines are routinely administered. Paradoxically, even in a country like Ecuador, which has higher reported rates of HPV vaccination than Colombia according to the WHO [45], one participant stated that she had never heard of the vaccine.

“*I wanted to say that my ignorance is even bigger because my children were born in Ecuador and there nobody talks about HPV (…). I was one of those mothers who said that I would not give my daughter that vaccine, until recently, when I went to get a booster for my other 10-year-old son and I spoke with a doctor who explained to me that it was absolutely safe, so I began to think differently*”.(mother of unvaccinated girl, 45 years old, private school)

Parents tend to trust in their doctors even when they have doubts about the vaccine. There is little reported discomfort with the vaccination-card requirement at school or with vaccination campaigns at schools, supporting the idea that institutions take care of people.

Quantitative data corroborated this finding, since 87.8% of respondent trusted the information they received about vaccines. We also found a bivariate relationship between the likelihood of getting vaccinated and receiving positive information about the vaccine (Spearman r = 0.338, *p* < 0.001). For parents, a binary logistic regression showed that the recommendation of the doctor was the most important factor related to the probability of the daughter getting vaccinated (See Table 4).

#### 3.2.3. Motivation

Regarding motivation, survey data revealed that some parents were concerned about the effectiveness, adverse effects, and safety of the vaccines. In focus groups, parents also revealed a conflict between their desire to protect their daughters from the harmful virus and their concerns about encouraging early sexual activity through vaccination. Often, this led to parents avoiding talking about the HPV vaccine.

##### The Vaccine as an Act of Care

Most parents considered that vaccination is a form of protection and care for their daughters, which leads them to demand clear information about its effectiveness and safety. “*I did it [vaccinate the daughter] first of all, to protect my daughter’s life*” (mother of girl with one dose, 41 years old, public school).

“*It is a matter of avoiding the disease; one does not know when it could happen, God willing it does not happen, but it is a way of caring for and protecting them*”.(mother of girl with one dose, 41 years old, private school)

Often, parents would mention the adverse event at Carmen de Bolívar, leading to uncertainty about the impact of the vaccine.

“*I have heard many mothers who did not want to vaccinate their daughters; I do not know why. A long time ago there was a problem with some vaccine, after which many mothers believe in these things, and they are scared to vaccinate their daughters*”.(mother of unvaccinated girl, 43 years old, private school)

##### Respect for the Personal Decision to Be Vaccinated

Some of the participants reported that vaccination is an individual act and at the same time, an act of collective responsibility. The decision cannot be forced, however. “*I think there must be free will there. Not only from parents, since at a certain age, young people make decisions regarding their bodies, and I think that it should not be mandatory*” (father of girl with one dose, 53 years old, private school).

Girls understood that to access to the vaccine they must go through two gatekeepers: first their parents and then their healthcare provider, acknowledging that, as minors, the decision rests with the parents. Some of the girls think that the vaccination is their decision; however, most of them think that this is a parental decision. “*Well, it depends, my mom talks about it with me; I tell her there is this vaccine, then she tells me yes, and the decision is mine, as she says: your body is yours*” (girl, one dose, 10 years old, public school).

“*I never had the power to decide if I wanted to get vaccinated or not; it was not a subject that I could get into*”.(Girl, one dose, 15 years old, public rural school)

##### Association of HPV with Sexuality

As HPV is a sexually transmitted infection, some parents state that prevention can be conducted through education in values, not talking about sexuality openly.

“*I consider that in the subject of the infection with the human papilloma virus, it is possible to take a look from the biological point of view as the subject of the disease; let’s say that it is detached from the subject of sexuality and that is perfectly well because one can approach the subject with one’s daughter like that*”.(Father of girl with one dose, 53 years old, private school)

Some parents stated that HPV infection occurred because people start sexual life early; they considered it an adult issue and did not consider their daughters susceptible to infection at their young ages. “*I have an acquaintance who had this cancer because she began her sexual life very young. That person passed away. She always warned us about the dangers of such a crazy sex life*” (mother of girl with one dose, 40 years old, public school)

This is also perceived by the girls, who state that their parents delegate the issue of sexuality to others, and that at school the information is provided superficially or not at all. “*Because it is a sexually transmitted disease, people avoid talking about it*” (girl with one dose, 15 years old, private school).

#### 3.2.4. The Impact of COVID-19 on HPV Hesitancy

As a result of the pandemic, people understand more about how vaccines work and what happens with a virus, and the analogy with COVID-19 allows them to understand that to be fully protected, they need several doses of a vaccine. Some of the young participants incorrectly think that the vaccines contain live viruses, however. In Colombia, greater understanding of COVID has allowed certain anti-vaccine discourses to be demystified (relatives of participants who were anti-vaccine but were now vaccinated) and negative ideas about vaccines to be discredited.

“*I don’t know exactly how the virus spreads, but I suppose that, like a normal virus, like when you get vaccinated against the coronavirus, you prevent yourself and others, that you don’t spread it, nor are you infected*”.(girl with one dose, 15 years old, public school)

Yet, especially in younger girls, different forms of HPV and COVID transmission, prevention, vaccination components and dosage, and side effects can be confused. “*Well, I think it is transmitted like when you cough on someone or if you are with that person a lot, the virus passes to you or something like that*” (unvaccinated girl, 9 years old, private school).

Importantly, regarding opportunities, the pandemic further limited access to the HPV vaccines and reduced the importance of health campaigns focused on HPV vaccines.

## 4. Discussion

The aim of this study was to understand the psychological predictors of HPV vaccine hesitancy in girls aged 15 and under and their parents/caregivers who live in and around Medellín. In this sample, more than half (57.1%) of participants have not initiated vaccination despite quantitative data showing that parents and caregivers who participated in the study perceived a high level of susceptibility to HPV and cervical cancer in girls and adolescents. They also had high awareness of the need for the vaccine. However, qualitative data revealed some unrealistic parental views of their daughters’ susceptibility to HPV; many parents thought that since their daughters are young, they were far from starting their sexual life. There is limited information about organized HPV vaccination programs in this sample, nor does it reveal how physical and social opportunities affect access to vaccines. Additionally, despite Colombia Health Ministry-led public health communications about the safety of the HPV vaccine, some participants remained concerned about the vaccine’s effectiveness and safety, and the side effects that the vaccine might cause.

Parents also were not especially aware of the vaccine’s benefits. This finding was also reported in an earlier study in a Colombian population [34]. Limited information and low vaccination rates are clear challenges for public health, given the prevalence of cervical cancer in Colombia and worldwide [2,5]. These findings point to the need for interventions that provide information about HPV, its relationship to cervical cancer, and the country’s national vaccination program to increase the public’s vaccine uptake. However, simply providing information does not cause a change in vaccination behaviour [46]. Perceived susceptibility to HPV is a fruitful target for future intervention approaches. Multiple studies have reported that perceived susceptibility is associated with a greater likelihood of getting vaccinated [47]. Educational interventions that also focus on susceptibility, by pointing out that HPV causes genital warts and that vaccination prevents diseases attributable to this virus have demonstrated effectiveness [48,49]. The WHO communication guidelines include educational interventions designed to increase HPV-vaccine uptake to increase the perception of susceptibility and the consequences of HPV infection [50].

The COVID-19 pandemic has highlighted the threat of an infectious disease and has increased awareness of the vaccine-development process, since many parents had never seen the devastating consequences of an infectious disease before 2020 [51]. This creates an opportunity to generate awareness about HPV and its relationship with cervical cancer, but also to promote the use of the vaccine as the best way of preventing future complications.

This study also showed how physical and social opportunities have an influence on vaccine-access behaviour. It is crucial that barriers to vaccine access be reduced, by offering inoculations at schools [52], at times when parents can give consent, since that is needed for vaccination. Educational institutions may be optimal settings for HPV-vaccine interventions, as natural gathering sites for students, parents, healthcare professionals, and teachers. School-based interventions may help to change attitudes and beliefs, as well as increase knowledge to increase HPV-vaccine uptake [53].

We found that people trust their healthcare professionals (HCPs) and the information they provide. This element is crucial when we consider that HCPs can promote vaccination by sharing accurate information and offering counselling to parents to facilitate decision making. Nonetheless, previous studies have found that the percentage of HCPs who speak with parents about HPV vaccines for their children is very low [54]. However, HCPs are critical to HPV-vaccine uptake, so provider-based interventions also offer considerable promise [55]. A provider-focus has also been recommended by the HPV Prevention and Control Board of Colombia, so broader implementation is a next step [56].

Lastly, it is important to remember the motivational component of this behaviour. Paradoxically, although most participants in this study trusted the vaccine information received and the healthcare system providing it, some of them were distrustful about vaccine safety, efficacy, and potential adverse effects. This aligns with studies in other countries, in which fears about side effects and vaccine safety, distrust in the vaccine, fear of the possibility of death, and negative comments from neighbours or acquaintances about the HPV vaccine were paramount in parents who did not want to vaccinate their children [57]. In Colombia, these fears were enhanced with the widespread coverage of the 2014 mass psychogenic response [26] that still remains in the public’s memories.

The thematic analysis showed that although the act of vaccine is seen as an individual decision, mothers have a crucial influence on their daughters. Generally, parents’ motivation was to protect and take care of their daughters. To increase the vaccination rates, parents should be involved, as has been suggested in other studies [36,48].

As HPV is primarily a sexually transmitted disease, the HPV vaccine is different from other vaccines for children. In Colombia, it is intended especially for girls [58], although HPV-vaccination should be gender neutral. Both men and women can develop cancer caused by HPV; further, it is easier to ensure a high coverage to achieve herd immunity when both sexes are vaccinated [59].

Several studies have reported that parents and caregivers may associate the vaccine with fears of compromising fertility [60] or giving children permission to become sexually active [61]. While the perceived risks are nearer term, the potential benefits of HPV vaccination are much longer term, since it takes 15 to 20 years for cervical cancer to develop in women with normal immune systems [62] and the estimated median time from HPV acquisition to cancer detection ranges from 17.5 to 26 years [63]. Educational interventions could highlight both the near-term and longer-term benefits of the HPV vaccine.

Creating opportunities for public discussion, led by trusted healthcare providers, for example, could help to allay these parental fears and increase HPV-vaccine uptake. Parents and educational institutions should be involved in the co-design of these public forums. Currently, governments should take clear actions to improve vaccination rates. Denmark shows how a nation grappled with negative media coverage of HPV vaccination with a parent information campaign designed to boost confidence in the safety of the vaccine [64]. Austria launched a renewed national discourse on vaccination, changing the message of vaccination from “save women’s lives” to “save lives” to disassociate the vaccine from gender [32].

The US Centers for Disease Control and Prevention (CDC) has recommended ten strategies for healthcare providers to increase the HPV vaccine. These range from giving active recommendations, recommending the HPV vaccine the same day and the same way as all other vaccines, preparing to answer more common questions, learning the reasons for vaccine refusal, and implementing systems to ensure that an opportunity to vaccinate is never missed [65].

Before concluding, we want to highlight the limitations of the study. It was conducted in the middle of the COVID-19-pandemic health emergency. This meant that all parent focus groups were remote, and their participation was limited by connection capability and speed. However, remote data-collection strategies may increase participation among those who may not otherwise be able to participate due to logistical or time constraints [66,67]. Use of a convenience sample from a geographically-limited set of participants may bias the findings and limit generalizability. For the quantitative analysis, we did not include variables that have predicted HPV-vaccine hesitancy in other studies, such as the level of parental education [68], so the findings may be confounded. Importantly, the use of mixed methods combines the strengths of both quantitative and qualitative research, by allowing the researcher to add insights that could be omitted when only a single method is adopted [69].

## 5. Conclusions

To summarize, this study improves the understanding of the capabilities, opportunities, and motivations associated with the HPV vaccine to reduce HPV-vaccine hesitancy for parents, girls, and adolescents in Medellín and nearby cities. It is important to provide information about the national vaccination program and HPV’s association with cervical cancer. Likewise, it is important to make vaccination access easier by returning to the strategy developed in 2012 to bring the program to schools, thereby reducing barriers to access. Lastly, parents’ concerns about vaccine safety and side effects should be acknowledged; healthcare leaders should create spaces for discussion so that the ideas behind the distrust can be dismantled.

## Figures and Tables

**Figure 1 vaccines-10-01187-f001:**
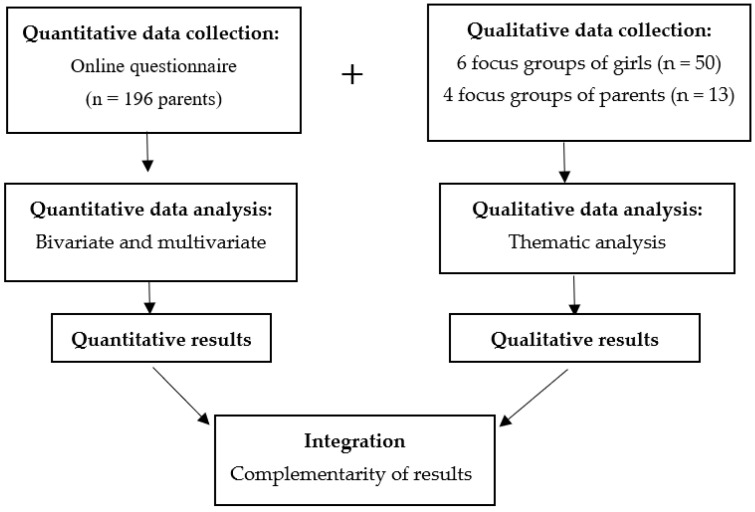
Convergent mixed-methods design.

**Table 1 vaccines-10-01187-t001:** Sociodemographic characteristics of online questionnaire participants (N = 196).

Adults	%	N
**Sex**		
Female	89.3%	175
Male	10.7%	21
**Socioeconomic status**		
Low	15.8%	31
Medium	23.5%	46
High	60.7%	119
**Daughter’s school type**		
Private	71.4%	140
Public	28.6%	56
**Doses**		
0	57.1%	112
1	42.9%	84
**Religious practice**		
Yes	82.7	162
No	17.3	34
**Healthcare-system affiliation**		
Contributory plan	96.9%	190
Subsidized plan	3.1%	6

**Table 2 vaccines-10-01187-t002:** Sociodemographic characteristics of girls and adolescents in focus groups (N = 50).

Girls and Adolescents	%	N
**Education level**		
Primary	60%	30
Secondary	40%	20
**Type of school**		
Private	48%	24
Public	52%	26
**School location**		
Urban	74%	37
Rural	26%	13
**Doses**		
0	42%	21
1	58%	29

**Table 3 vaccines-10-01187-t003:** A joint display of factors influencing HPV-vaccine hesitancy to parents/caregivers and daughters based on the COM-B model.

Source of Behaviour	Quantitative Data from the Online Questionnaire	Themes from the Focus Groups
Capability	11.7% did not perceive that their daughter would be susceptible to contracting HPV. Relationship between likelihood of vaccinating daughter and perceived susceptibility to HPV; Spearman r = 0.309, *p* < 0.001. 84.7% perceived that their daughter was susceptible to cervical cancer. 96.9% were aware that they should get their daughter vaccinated. 69.4% did not receive any information about HPV vaccination for their daughter. 87.8% trusted the information they received about vaccines. Relationship between likelihood of getting vaccinated and receiving positive information about the vaccine; Spearman r = 0.338, *p* < 0.001	Lack of information Relationship between HPV and cervical cancer
Opportunity	Relationship between likelihood of getting vaccinated and medical recommendation; Spearman r = 0.221, *p* < 0.01 Relationship between likelihood of getting vaccinated and seeing others get vaccinated; Spearman r = 0.158, *p* < 0.05	Trust in traditional institutions
Motivation	30.1% were concerned about vaccine effectiveness 32.7% were concerned because vaccines may have adverse effects 31.6% were concerned about vaccine safety Relationship between likelihood of getting vaccinated and vaccine safety; Spearman r = 0.277, *p* < 0.001.	Association of HPV with sexuality The vaccine as an act of care Respect for the personal decision to be vaccinated

**Table 4 vaccines-10-01187-t004:** Logistic regression results for predicting the reasons to vaccinate daughters using socioeconomic position (SEP), religion, and other influences as independent variables.

Step	Variable Entered	B	Wald	Sig	Exp (B)	C.I for Exp (B)	
						Lower	Upper
1	SEP medium	−0.575	0.616	0.432	0.562	0.134	2.366
	SEP high	−0.736	1.424	0.233	0.479	0.143	1.604
	Religion	−0.532	0.828	0.363	0.587	0.187	1.847
2	Cost	−0.065	0.045	0.832	0.937	0.514	1.709
	Easy access to vaccine	0.142	0.245	0.620	1.153	0.657	2.025
	Social norms	−0.053	0.017	0.896	0.949	0.431	2.088
	Medical recommendation	−0.713	5.167	0.023	0.490	0.265	0.906
	Others’ recommendations	−0.593	2.408	0.121	0.553	0.261	1.169
	Safety	0.873	3.200	0.074	2.394	0.920	6.233
	Susceptibility	−0.584	3.617	0.057	0.558	0.306	1.018
	Positive information	−0.460	2.365	0.124	0.631	0.351	1.135

Note. SEP = Socioeconomic position.

## Data Availability

The data supporting reported results can be found at https://osf.io/n396v/?view_only=6f04d370460a419bbeceba5600ce91c3 (accessed on 31 May 2022).

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
