# Peer review of "HPV-Vaccine Hesitancy in Colombia: A Mixed-Methods Study"

_vaccines, 2022, doi:10.3390/vaccines10081187_

Round 1

Reviewer 1 Report

This an interesting paper . This work is well described and no changes are necessary

Author Response

Response to Reviewer 1 Comments

Point 1: This an interesting paper. This work is well described, and no changes are necessary

Point 2: Moderate English changes required

Response: Thanks for reviewing our paper and your comments. We have made the moderate English language changes that have been requested.

Reviewer 2 Report

The manuscript titled “HPV Vaccine hesitancy in Colombia: A Mixed Methods Study” is novel as it is the first to explore the factors influencing HPV vaccine hesitancy alongside the COVID vaccine within an integrative model of behavior change, the Capability-Opportunity-Motivation-Behavior (COM-B) model. The study group found that Covid-19 has had an impact on the awareness of HPV and HPV vaccination. The group outlines that lack of information about the vaccination programs, concerns about vaccine safety and the relationship between HPV and sexuality could be related to vaccine hesitancy. This is an important field of study and provides useful information about vaccine hesitancy. This manuscript has been written extremely well and I would recommend accepting the manuscript with minor revisions.

Strengths:

The aim of this study was to understand the psychological predictors of HPV vaccine hesitancy in girls aged 15 and under and their parents/caregivers who live in and around 369 Medellin. In this sample more than half (57.1%) of participants have not initiated vaccination despite quantitative data showing that parents and caregivers who participated in the study perceive a high level of susceptibility to HPV and cervical cancer in girls and adolescents and have high awareness of the need for the vaccine. To summarize, this study improves understanding of the capabilities, opportunities and motivations associated with HPV vaccine to reduce hesitancy from parents, girls, and adolescents in Medellín and nearby cities.

Limitations:

It was conducted in the middle of the COVID-19 pandemic health emergency. This meant that all parental focus groups were remote, and their participation was limited by connection capability and speed.

Use of a convenience sample from a geographically-limited set of participants may bias the findings, and limit generalizability. For the quantitative analysis, we did not include variables that have predicted HPV vaccine hesitancy in other studies, such as level of parental education.

Minor Comments:

The Sample size for the study was small.

An important piece of information that is missing from the manuscript is the socioeconomic and educational status of the parents. This might have considerable influence on vaccine hesitancy among participants.

Author Response

Response to Reviewer 2 Comments

Point 1: The Sample size for the study was small.

An important piece of information that is missing from the manuscript is the socioeconomic and educational status of the parents. This might have considerable influence on vaccine hesitancy among participants.

Response 1: Thank for your helpful comment. In the limitations, we acknowledged the small sample size since the study was conducted in the middle of the COVID-19 pandemic health emergency. This meant that all parental focus groups were remote, and their participation was limited by connection capability and speed. More importantly, however, this was a mixed methods study, with qualitative components that sought data saturation, yielding a smaller sample size. The strength of the mixed methods is the systematic integration of the qualitative and quantitative data. We conducted a power analysis for the cross-sectional survey and found it reasonable, given the prevalence rates of HPV vaccination extant in Colombia.  

As noted by the reviewer, and as a limitation of the paper, we were not able to collect the educational status of the parents.

Reviewer 3 Report

This paper talks about HPV vaccine hesitancy among girls in one city. Unlike wet lab research works, this study is designed based on the views of people like daughters and parents and is analyzed scientifically. It’s very interesting to see how people react about a scientific product, in this case, the HPV vaccine. Currently, we have so many lab-based HPV studies.

However, there are not enough studies talk about how people understand the relationship between HPV and cervical cancer, how parents’ ideas, schools, and other physical and social factors impact HPV vaccination and how trustable the HPV vaccines are among people in a certain area.

So, this paper has its significance for sure, and it would be cool to know how HPV vaccinations are considered in other cities that are completely different than the city in this study. It would also be cool to see how males think about taking HPV vaccines themselves. Overall, the paper is well delineated and organized.

Some small points need to be checked: 1. Lane 42-48 needs citations; 2. Please watch the formatting. Some words from the interviewee are italic, however, some words from the interviewee are not. 

Author Response

Response to Reviewer 3 Comments

Point 1: This paper talks about HPV vaccine hesitancy among girls in one city. Unlike wet lab research works, this study is designed based on the views of people like daughters and parents and is analyzed scientifically. It’s very interesting to see how people react about a scientific product, in this case, the HPV vaccine. Currently, we have so many lab-based HPV studies.

However, there are not enough studies talk about how people understand the relationship between HPV and cervical cancer, how parents’ ideas, schools, and other physical and social factors impact HPV vaccination and how trustable the HPV vaccines are among people in a certain area.

So, this paper has its significance for sure, and it would be cool to know how HPV vaccinations are considered in other cities that are completely different than the city in this study. It would also be cool to see how males think about taking HPV vaccines themselves. Overall, the paper is well delineated and organized.

Some small points need to be checked: 1. Lane 42-48 needs citations; 2. Please watch the formatting. Some words from the interviewee are italic, however, some words from the interviewee are not. 

Response 1: Thank you for your helpful comments. We also expect to compare different cities in Colombia in future studies because of cultural differences within Colombia. Medellín is an excellent site for this study, however, as the second-largest city in Colombia, after Bogotá, and the capital of the department of Antioquia. With its surrounding area that includes nine other cities, Medellín has a population of more than 4 million people. The city houses our universities, major academic centers.  We have developed relationships with the schools in many parts of the city, facilitating research.

The citation required for the lines 42-48 was updated and the formatting of the paper was reviewed.

Reviewer 4 Report

In Colombia the uptake of HPV vaccine rate dropped from 96.7% after its introduction in 2013 to 9% in 2020. Cordoba-Sanchez et al. have conducted a convergent mixed methods study to identify the behavioral components of HPV vaccine hesitancy in females aged 15 and under, and their families. Lack of information about the vaccination programs, concerns about vaccine safety and the relationship between HPV and sexuality could be related to vaccine hesitancy. In conclusion, trust in medical recommendations, and campaigns focused on the idea that vaccination is a way of protecting daughters from cervical cancer could improve HPV vaccine uptake.

The claims are properly placed in the context of the previous literature. The experimental data support the claims. The manuscript is written clearly enough that most of it is understandable to non-specialists. The authors have provided adequate proof for their claims, without overselling them. The authors have treated the previous literature fairly. The paper offers enough details of methodology so that the experiments could be reproduced.

Comments

Several countries have experienced reduced vaccine coverage of the HPV vaccine as a result of unfortunate media coverage. This should be included both in the introduction and in the discussion.

In 2014, Denmark experienced a rapid decline in vaccination uptake for the human papillomavirus (HPV) vaccine after a successful introduction of the vaccine in 2009. Before the decline, the uptake of the first HPV vaccine was around 90% for girls born in the period 1998 to 2000, while it dropped to 54% for girls born in 2003. The decline followed negative public attention from 2013 coinciding with increasing suspected adverse-event reporting to the Danish Medicines Agency.  

Suppli, C.H., Hansen, N.D., Rasmussen, M. et al. Decline in HPV-vaccination uptake in Denmark – the association between HPV-related media coverage and HPV-vaccination. BMC Public Health 18, 1360 (2018). https://doi.org/10.1186/s12889-018-6268-x

The strong negative media coverage and the powerful social media, where stories of the supposedly harmful effects of HPV vaccination were widely shared, overshadowed the scientific evidence documenting the safety and effectiveness of the HPV vaccine. In 2015, the European Medicines Agency (EMA) and the World Health Organization (WHO) reported no evidence of any HPV vaccine safety issues and the Danish Health Authority supported the risk/benefit ratio of the vaccine in a public statement.

Despite these reports and a starting increase in vaccination coverage, rebuilding confidence in the safety of the HPV vaccine and restoring vaccine coverage still required extraordinary efforts. The Danish Cancer Society established a telephone hotline, where parents could call with questions regarding HPV vaccination, and in May 2017, an information campaign ‘‘Stop HPV - stop cervical cancer’’ was initiated. Before designing the campaign, a survey was conducted to investigate parents’ concerns about HPV vaccination and some of the main findings were that the parents requested more information on the HPV vaccine and that their primary source of information was online media, particularly Facebook. The information campaign was a collaboration between the Danish Health Authority, the Danish Cancer Society, and the Danish Medical Association. The campaign used a website and a Facebook page to post different types of information, case stories, and short movies. Thereby it was possible to communicate directly with the target group. In addition, an effort was made to communicate the scientific results related to the effect of HPV vaccination, where Denmark was one of the first countries to report a decreased risk of high-grade cervical lesions among vaccinated women.

https://www.hpvworld.com/articles/hpv-vaccination-crisis-and-recovery-the-danish-case/

The experience in Denmark offers one of the first opportunities to document how a nation grappled with negative media coverage of HPV vaccination and the steadying impact of action by national authorities.

Hansen PR, Schmidtblaicher M, Brewer NT. Resilience of HPV vaccine uptake in Denmark: Decline and recovery. Vaccine. 2020 Feb 11;38(7):1842-1848. doi: 10.1016/j.vaccine.2019.12.019. Epub 2020 Jan 7. PMID: 31918860.

https://pubmed.ncbi.nlm.nih.gov/31918860/

On Nov 26, 2021, the Ministry of Health, Labour, and Welfare of Japan officially issued an announcement to resume active recommendations of the human papillomavirus (HPV) vaccine, which had been suspended since June, 2013.1 The new announcement now clearly advises municipalities to recommend the vaccine in accordance with Article 8 of the National Immunization Law, which includes sending notifications and vouchers individually to the target population (ie, girls aged 12–16 years).1 Municipalities are expected to restart such active recommendations from April, 2022.

Even during the period when active recommendations were suspended, the HPV vaccine was maintained as a part of the national immunisation programme and provided free to girls in the target population seeking vaccination.2 However, the target girls were not individually notified that they could have the vaccine. This situation continued despite large-scale epidemiological studies showing the effectiveness and safety of the vaccine in Japan and worldwide, and the scientific community repeatedly calling for the resumption of active recommendations by the Japanese government.3,  4,  5 As a result, public resistance regarding the HPV vaccine in Japan has remained, and vaccination coverage stagnated at a low rate (<1%) over the past 7 years.

Haruyama R, Obara H, Fujita N. Japan resumes active recommendations of HPV vaccine after 8·5 years of suspension. Lancet Oncol. 2022 Feb;23(2):197-198. doi: 10.1016/S1470-2045(22)00002-X. PMID: 35114115.

https://pubmed.ncbi.nlm.nih.gov/35114115/

The HPV vaccine Gardasil was licensed comparatively early in Austria, notably for girls and boys, in October 2006, against a rather particular cervical cancer screening infrastructure that warrants explanation. In Austria, a nation-wide cervical cancer screening strategy was first set up in 1970. Women above the age of 20, or at the latest two years after commencing sexual activity, are offered annual screenings, though remarkable variation exists even within Austria (e.g. Krebshilfe, 2014, Hauptverband, 2014). Despite the early introduction of the program, it has remained flawed due to its loose and opportunistic character: participation depends on individual initiative rather than a national recall system. Estimates suggest an annual screening participation of around 30%

The inexplicable death of a young girl in October 2007 in Upper Austria only a few weeks after her first HPV vaccination dose added further fuel to the emerging debate (cf. Stöckl, 2010, Interview 1, 7). There was a risk that historical skepticism towards vaccination (particularly against MMR) in sub-communities would gain new momentum. While empirical data regarding this skepticism is limited, most respondents refer to it as a given. Notably, in Austria, anti-vaccination sentiment is not primarily religiously driven, but it includes views of vaccinations as interventions in the natural course of childhood, fears of “unnatural” adjuvant substances, such as aluminum (Interview 1), and, as a recent survey suggests, widespread beliefs that vaccination may cause allergies (OTS, 2013).

Vaccination against the sexually transmitted Human Papilloma Virus (HPV), a necessary agent for the development of cervical cancer, has triggered much debate. In Austria, HPV policy turned from “lagging behind” in 2008 into “Europe's frontrunner” by 2013. Drawing on qualitative research, the article shows how the vaccine was transformed and made “good enough” over the course of five years. By means of tinkering and shifting storylines, policy officials and experts disassociated the vaccine from gender, vaccine manufacturers, and youth sexuality. Ultimately, the HPV vaccine functioned to strengthen the national immunization program. To this end, preventing an effective problematisation of the extant screening program was essential.

Paul KT. "Saving lives": Adapting and adopting Human Papilloma Virus (HPV) vaccination in Austria. Soc Sci Med. 2016 Mar;153:193-200. doi: 10.1016/j.socscimed.2016.02.006. Epub 2016 Feb 6. PMID: 26921834; PMCID: PMC4789483.

https://pubmed.ncbi.nlm.nih.gov/26921834/

Top 10 Tips for HPV Vaccination Success: Attain and Maintain High HPV Vaccination Rates

https://www.cdc.gov/hpv/hcp/2-dose/top-10-vaxsuccess.html

How to ensure high coverage of the HPV vaccine

1. The HPV-vaccination should be school based.

2. The HPV-vaccination should be free of charge.

3. The HPV-vaccination should start at 9 years old.

4. The HPV-vaccination should be gender neutral (both boys and girls).

5. It is better to use “opt-out” for parents who refuse vaccination their children instead of “opt-in” (if the parents do not sign against vaccination on the consent form, the child should receive vaccine).

6. Parents who want to opt-out should receive medical counseling.

7. The HPV-vaccine should be co-administered together with other vaccines in the vaccination program. The best way is to use combined vaccines for example meningitis, HPV and whooping cough in the same shot.

8. Taking the HPV vaccine or not should not be a big personal choice for every single child, or parent. The HPV vaccine is recommended by the health authorities, just take it.

Minor revisions

Line 26, “Cervical cancer is the 3rd most common cancer among women worldwide” => “Cervical cancer is the fourth most common cancer among women globally”

https://www.who.int/news-room/fact-sheets/detail/cervical-cancer

Line 29, “In Colombia, cervical cancer has an incidence of 18.3” => "In Colombia, cervical cancer has a crude incidence rate of 18.3 per 100,000 women per year"

https://hpvcentre.net/statistics/reports/COL.pdf?t=1624248056406

In 2008, Cervical cancer in Colombia was the first cause of cancer mortality and the second cause of cancer incidence among women. The highest mortality rates are observed in the most deprived regions ( along the main rivers, harbors, cities in the country borders). In 2008 a total of 4,736 new cases and 2,154 deaths were estimated to have occurred. These numbers correspond to an age-adjusted incidence rate of 21.5 per 100,000 and a mortality rate of 10.0 per 100,000).

Muñoz N, Bravo LE. Epidemiology of cervical cancer in Colombia. Colomb Med (Cali). 2012 Dec 30;43(4):298-304. PMID: 24893303; PMCID: PMC4001966.

https://pubmed.ncbi.nlm.nih.gov/24893303/

Line 431-433, “As HPV is a sexually transmitted disease, the HPV vaccine is different from other vaccines for children because, in Colombia, it is intended especially for girls, although it can also be beneficial for boys”

HPV-vaccination should be gender neutral. Both men and women can develop cancer caused by HPV. It is easier to ensure a high coverage when vaccinating both boys and girls. It is easier to achieve herd immunity when both sexes are vaccinated. The vaccine program becomes more robust for fluctuations in the coverage of the HPV vaccine when both sexes are vaccinated.

Line 436-437, “since a woman might show symptoms of cervical cancer up to 10 years after getting infected” => “It takes 15 to 20 years for cervical cancer to develop in women with normal immune systems”

https://www.who.int/news-room/fact-sheets/detail/cervical-cancer

The natural history of human papillomavirus (HPV)-induced cervical cancer (CC) is not directly observable, yet the age of HPV acquisition and duration of preclinical disease (dwell time) influences the effectiveness of alternative preventive policies. We performed a Cancer Intervention and Surveillance Modeling Network (CISNET) comparative modeling analysis to characterize the age of acquisition of cancer-causing HPV infections and implied dwell times for distinct phases of cervical carcinogenesis. The median time from HPV acquisition to cancer detection ranged from 17.5 to 26.0 years across the four models.

Burger EA, de Kok IMCM, Groene E, Killen J, Canfell K, Kulasingam S, Kuntz KM, Matthijsse S, Regan C, Simms KT, Smith MA, Sy S, Alarid-Escudero F, Vaidyanathan V, van Ballegooijen M, Kim JJ. Estimating the Natural History of Cervical Carcinogenesis Using Simulation Models: A CISNET Comparative Analysis. J Natl Cancer Inst. 2020 Sep 1;112(9):955-963. doi: 10.1093/jnci/djz227. PMID: 31821501; PMCID: PMC7492768.

https://pubmed.ncbi.nlm.nih.gov/31821501/

Author Response

Response to Reviewer 4 Comments

Point 1:  Introduction can be improved

Cited references can be improved

Response 1: Thank you; we have enriched the introduction. All the suggested references to Japan, Denmark and Austria are now included in the introduction and in the discussion. 

Point 2:  Minor revisions

Line 26, “Cervical cancer is the 3rd most common cancer among women worldwide” => “Cervical cancer is the fourth most common cancer among women globally”

 Line 29, “In Colombia, cervical cancer has an incidence of 18.3” => "In Colombia, cervical cancer has a crude incidence rate of 18.3 per 100,000 women per year"

Line 431-433, “As HPV is a sexually transmitted disease, the HPV vaccine is different from other vaccines for children because, in Colombia, it is intended especially for girls, although it can also be beneficial for boys”

Line 436-437, “since a woman might show symptoms of cervical cancer up to 10 years after getting infected” => “It takes 15 to 20 years for cervical cancer to develop in women with normal immune systems”

Response 2: All the suggested modifications have been made on the indicated lines.